# Evaluation Methods and Measurement Challenges for Industrial Exoskeletons

**DOI:** 10.3390/s23125604

**Published:** 2023-06-15

**Authors:** Ya-Shian Li-Baboud, Ann Virts, Roger Bostelman, Soocheol Yoon, Amaan Rahman, Lucia Rhode, Nishat Ahmed, Mili Shah

**Affiliations:** 1National Institute of Standards and Technology, Gaithersburg, MD 20899, USA; ann.virts@nist.gov (A.V.);; 2Smart HLPR LLC, Troutman, NC 28166, USA; 3Institute for Soft Matter, Georgetown University, Washington, DC 20057, USA; 4Department of Electrical Engineering, Albert Nerken School of Engineering, The Cooper Union for the Advancement of Science and Art, New York, NY 10003, USA; 5Department of Mathematics, Albert Nerken School of Engineering, The Cooper Union for the Advancement of Science and Art, New York, NY 10003, USA

**Keywords:** exoskeleton, evaluation methods, performance metrics

## Abstract

In recent years, exoskeleton test methods for industrial exoskeletons have evolved to include simulated laboratory and field environments. Physiological, kinematic, and kinetic metrics, as well as subjective surveys, are used to evaluate exoskeleton usability. In particular, exoskeleton fit and usability can also impact the safety of exoskeletons and their effectiveness at reducing musculoskeletal injuries. This paper surveys the state of the art in measurement methods applied to exoskeleton evaluation. A notional classification of the metrics based on exoskeleton fit, task efficiency, comfort, mobility, and balance is proposed. In addition, the paper describes the test and measurement methods used in supporting the development of exoskeleton and exosuit evaluation methods to assess their fit, usability, and effectiveness in industrial tasks such as peg in hole, load align, and applied force. Finally, the paper includes a discussion of how the metrics can be applied towards a systematic evaluation of industrial exoskeletons, current measurement challenges, and future research directions.

## 1. Introduction

According to the U.S. Bureau of Labor Statistics, the retail, manufacturing, as well as healthcare and social services, industries accounted for 50 percent of all the work-related musculoskeletal disorders (WMSDs) in 2018 that involved days away from work [1]. Laborers and freight, stock, and material movers were the occupations with the largest number of reported WMSDs [1]. Similarly, as much as 30 percent of all lower back pain can be attributed to occupational risk factors [2]. Such risk factors in manufacturing include awkward postures, hand force, and material handling of objects. Awkward postures include, but are not limited to, twisting, bending, reaching, overhead work, or where the back or neck is bent over 30 degrees without support [3]. In 2022, over exertion due to object handling and awkward postures were among the top ten causes of non-fatal workplace injuries and have an estimated economic cost of USD 16.64 billion in the U.S. [4]. When engineering controls are unavailable, industrial exoskeletons have the potential to reduce workplace injuries by augmenting, reinforcing, or amplifying part of a worker’s musculoskeletal system in automotive [5,6,7,8] and steel [9] manufacturing. In the automotive industry, final assembly involves fast-paced, high-precision tasks requiring manual intervention for doors, underbodies, and car interiors, where workers are frequently in awkward postures [8]. Industrial exoskeletons are designed to support the workers by effectively transferring the forces from repetitive and sustained motions to improve task ergonomics. Effective application of industrial exoskeletons can benefit from standard evaluation methods to understand the capabilities and limitations of an exoskeleton depending on the user and the task characteristics [10,11].

While exoskeletons have demonstrated potential for reducing musculoskeletal injuries and improving worker productivity, a standard framework for evaluating exoskeleton task performance, mobility, and safety can help both manufacturers and potential users to apply the technology effectively to industrial tasks.

The types of exoskeleton evaluation methods can be categorized as (a) in vitro, where evaluation is performed in a controlled environment on a physical model, such as a mannequin; (b) in vivo, where evaluation is performed with human test subjects either in a laboratory environment or an industrial environment; and (c) in silico, where computational simulations are applied to analyze human–exoskeleton interactions [12]. This study focuses on in vivo exoskeleton test methods. Industrial exoskeleton evaluations can be conducted in simulated environments in a laboratory, simulated field environments, and field trials where workers perform industrial tasks. While laboratory studies provide access to the instrumentation needed for quantitative measurements necessary for scientific understanding of the impact of industrial exoskeletons for potential use cases, field studies can be directly generalized to determine an exoskeleton’s implementation effectiveness from the perspective of expert workers and intended final work contexts [8].

The International Organization for Standardization’s (ISO)/Technical Committee (TC) 299 Robotics [13], Working Groups 2 and 4 develop safety and performance standards for service robots which include active (i.e., powered) exoskeletons. Working Group 1 develops vocabulary for robots, including terms associated with exoskeletons. ISO/TC 299/Joint Working Group (JWG) 5 is a collaboration between ISO/TC 299 and International Electrotechnical Commission/Subcommittee (IEC/SC) 62: Medical robot safety, which includes the safety of robotic exoskeletons used for medical purposes, such as patient rehabilitation. ISO 13482:2014 specifies safety requirements for personal-care robots and the need for risk assessment and hazard identification [14].

ASTM Committee F48 on Exoskeletons and Exosuits [15] has been tasked to develop performance standards for exoskeletons. In support of the development of standard industrial-exoskeleton evaluation methods for manufacturing, the National Institute of Standards and Technology (NIST) has developed test methods and provided data sets capturing the motion of users with exoskeletons using video cameras, depth perception sensors, physiological sensors (such as heart rate monitors), productivity data (such as task rate), as well as subjective assessments on the overall comfort and effectiveness of industrial exoskeletons. The goal of this study is to provide a survey of standard and proposed industrial-exoskeleton evaluation methods. Standard test methods typically include the flexible implementation of test procedures and metrics that depend on the intended task and purpose of the exoskeleton. For evaluating exoskeleton performance, both quantitative and qualitative measurements on the exoskeleton user or the exoskeleton can be applied with the goal of moving towards non-invasive quantitative metrics in field studies. Therefore, this study also provides a review of metrics for exoskeleton performance evaluation and discusses some of the measurement challenges in providing effective evaluations.

## 2. Materials and Methods

NIST conducted a series of exoskeleton test method development studies with over 100 subjects performing simulated industrial manufacturing tasks with and without an exoskeleton. The studies were approved by the NIST Institutional Review Board (IRB). The test methods developed used the Position and Load Test Apparatus for Exoskeletons (PoLoTAE), shown in Figure 1, which is a simulated framework to evaluate a variety of common manufacturing tasks including load handling [16], peg in hole [17], applied force, and load alignment.

In addition, a set of novel optical tracking marker artifacts were worn by the subjects for synchronous tracking of exoskeleton and human leg position and orientation using an optical tracking system (OTS). The standard test artifacts were intended to address the challenges of variation in measurement uncertainty between different marker clusters, marker movement on soft tissue, and marker occlusion when using traditional biomechanical marker models while wearing an exoskeleton. Kinematic misalignment between the user and the exoskeleton can also have performance and safety implications, and tests have been performed to synchronously track the exoskeleton and human lower limb position and orientation to evaluate exoskeleton fit to the user based on the joint angle difference between the exoskeleton and the test subject, in degrees [18].

The study data has been published [19] and includes data from 68 subjects (59 percent of total subjects) after receiving publication consent. The data collected includes heart rate, motion capture, videos, skeletal-joint tracking, and survey data. The data collected progressed chronologically in types, quantity, and measurement location as new instrumentation and measurement methods were integrated. The data set was provided to allow researchers to further advance exoskeleton safety and performance metrology methods, analysis techniques, and improved test methods. The data set has provided a means to develop training sets for human–exoskeleton joint identification toward low-cost field-deployable measurement methods using monocular cameras [20].

The test development studies provided insights into the voluntary consensus standard test development process under ASTM Committee F48 on Exoskeletons and Exosuits. The current and planned test methods and associated information standards are provided to enable potential exoskeleton manufacturers and users to evaluate an exoskeleton’s ability to support specific industrial tasks. Through the IRB study and standards development, a notional classification of standard and future standard test methods and metrics is proposed. The classification also includes additional measures from literature reviews and site visits. The objective of this study is to guide potential exoskeleton users to readily select the relevant standard test methods and metrics for their industrial tasks.

## 3. Towards Exoskeleton Test Methods and Standards

Industrial stakeholders recognize the need for standards in exoskeleton safety, quality, performance, ergonomics, and terminology for systems and components during the full life cycle of the product. This life cycle encompasses the time prior to usage, maintenance, and disposal with considerations for information-technology security. ASTM Committee F48 includes standardization activities covering industrial, emergency response, medical, military, and consumer applications covering passive and active systems, as well as strength or mobility-enhancing and load-relieving systems that reduce strain and fatigue on muscles and joints. Exoskeletons can be designed to enhance joint mechanics [21], providing an enhancing effect, or can be designed to reduce joint torque [22].

### 3.1. Information Standards

Table 1 enumerates current and proposed industrial-exoskeleton information standards. Information standards provide the data and description of the user, exoskeleton, task, and environmental contexts to support reproducible and repeatable tests. Each type of exoskeleton evaluation method supports the collection of different sets of parameters such as the test apparatus, exoskeleton, user, and environment. Several key factors can impact exoskeleton performance. The series of information standards including user information, exoskeleton configuration, exoskeleton-user training, and exoskeleton fit to user for each exoskeleton test trial is intended to improve the repeatability of exoskeleton performance testing. Exoskeletons may provide different benefits based on user condition, demographics, and exoskeleton configuration, as well as the ability to properly fit a user’s anthropometric measurements. Proper training and fit also improve safety where the user understands the intended use and where proper fit enables the exoskeleton to move synchronously with the user to maintain balance with predictable forces and kinematics.

### 3.2. Evaluation Standards

Table 2 extends the contextual information to include the evaluation of exoskeletons for specific industrial tasks. These evaluation standards are categorized into task-based and mobility-based evaluation standards. The task-based evaluation standards and proposed standards include load handling [31] and applied force tasks such as grinding, sanding, and overhead painting. Several exoskeleton performance standards intended to evaluate the effects of exoskeletons on the user’s mobility include sit to stand [32], gait [33], horizontal confined spaces [34], beams [35], gaps [36], and hurdles [37]. The exoskeleton evaluation standards provide a framework in which an evaluator can decide on the specific environment, measurements, test procedures, and information collection based on the intended task and the intended use of the exoskeleton. For both the safety and effectiveness of using an exoskeleton, evaluators can consult with the exoskeleton manufacturer to determine whether the test protocol follows proper exoskeleton maintenance, operations, and safety risk management. In addition, the necessary training for both the evaluators and the test subjects is sufficient to safely operate the exoskeleton and follow the test protocol.

One common industrial task involves load and material handling. Back exoskeletons are intended to reduce mechanical loading on the spine and strains on the back muscles. Studies have demonstrated the need for both force and electromyography (EMG) measurements as reductions in spine loading are not always correlated [47]. EMG sensors, for the neck, back, shoulder, arm, and leg muscles, have been applied to track muscle activity. EMG signals provide quantitative measures to better understand the impact of an exoskeleton on physiological exertion. EMG signals can track the magnitude as well as the temporal and spatial properties of the physiological changes introduced with the use of exoskeletons [48]. The amplitude of the EMG signal describes the peak muscle activity. Typically, a decrease in peak muscle activity is expected with the use of exoskeletons. An increase in peak muscle activity can imply the transfer of loads from one muscle region to another. A decrease in the mean power frequency of EMG signals has also been shown to indicate neuromuscular fatigue [49,50] and to indicate the point before and after the force begins to decline, which is also known as the endurance point [51].

Evaluations can also benefit from capturing user and exoskeleton kinematics to observe fit as well as temporal and spatial changes in motions when wearing an exoskeleton compared to a baseline condition. Videos and OTS can be used to track user kinematics for human and exoskeleton motion analysis [18,52]. An OTS can be used by modifying the marker models to manage occlusions. The marker models are a set of OTS markers placed precisely on anatomical landmarks to track the exoskeleton wearer’s motion. The tracking software can include baseline and conventional marker sets [53]. The advantage of using clinically validated marker models, such as the Helen Hayes marker set [54], is that the measured data can be used by biomechanical analysis software to derive more repeatable and accurate joint angle measurements. In addition, the kinematic data from tracking the exoskeleton wearer’s marker set can be combined with kinetic measurements, such as force plates, to compute joint moment profiles [54]. The challenge in using an OTS to track an exoskeleton wearer’s motion is primarily marker occlusion. To overcome occlusion, slight modifications to the marker model have been applied to track the wearer’s motion. Another method is to use common artifacts with a set of markers both on the exoskeleton and on the wearer to synchronously track the kinematic alignment [18]. Markerless human motion analysis methods, using video or depth-perception cameras, rely on stereophotogrammetric methods to estimate bone position, bone orientation, and joints within a rigid body framework [55,56] with an associated uncertainty [57]. Video-based methods have the advantage of being portable, low cost, and readily adaptable for use in field environments.

Inertial measurement units (IMUs) can also be used to manage occlusions and improve evaluation flexibility in industrial field studies. The IMU accuracy is limited by the calibration process, subject to the wearer’s precision in performing the postures necessary for joint-axis identification [58]. A 3° root mean square error between OTS and IMU measurements was observed in one study. IMUs have been used to detect the onset of movements as well as the classification of task techniques. For example, one study used IMUs to classify different lifting techniques such as squatting, stooping, and asymmetric lifting [59].

Industrial tasks involving humans also tend to require dynamic and versatile motions. Therefore, mobility tests have also been developed. Mobility test methods while using exoskeletons include, but are not limited to, movements, with and without loads, climbing, and agility [60].

The evaluation methods described in Table 1 and Table 2, along with academic research, have explored the use of several metrics used to evaluate exoskeletons. A comprehensive measurement approach covering an exoskeleton’s function (Table 3), ergonomic effects (Table 4), task performance effects (Table 5), and usability (Table 6) have been considered in evaluations of industrial exoskeletons for industrial adoption. Quantitative measurements of the user with and without an exoskeleton can be categorized into kinematic, kinetic, and physiological. Measurement analysis typically focuses on the ability of the exoskeleton to reduce physiological cost and to minimize forces that can lead to or exacerbate musculoskeletal injuries. Kinematic alignment during dynamic motions has also been evaluated as a fit metric [18].

Metabolic costs of exoskeleton use due to the additional load can offset the benefits of the exoskeleton. Measures to evaluate the metabolic costs based on physiological measures such as oxygen consumption and heart rate as well as metabolic cost models. Locomotory economy metrics, such as cost of transport (COT), are performance measures evaluating the metabolic cost efficiency of an exoskeleton as the wearer, with or without additional load, moves from one location to another. For example, for exoskeletons designed to augment load-carrying abilities by reducing the work to the wearer, COT can be derived by measuring the rate of oxygen consumption of the subject with and with the exoskeleton [62]. One metabolic cost model, the Augmentation factor, approximates the costs associated with the added device weight requiring a net energy to the user, with the augmented effects of providing the user with positive mechanical power [65].

In order to maximize the benefits of exoskeletons, users can select the required augmentations according to the intended tasks. Exoskeletons are designed to function for specific tasks which inform their designs. Exoskeletons can offer different types of degrees of freedom (DoF) which are assisted (actuated) or non-assisted (non-actuated). The kinematic structure also varies from soft exosuits to rigid anthropomorphic and non-anthropomorphic exoskeletons. In addition, based on the actuation mechanism, an exoskeleton can further be categorized into passive, quasi-passive, quasi-active, and active exoskeletons. Passive exoskeletons rely only on mechanical components, such as springs and dampers. Quasi-passive exoskeletons leverage passive components for assisting human joints and augmenting with electronic sensors (including force, IMU, and EMG sensors) to engage or disengage the passive components [92]. Quasi-active exoskeletons use both passive and active joints to gauge user intent and provide active support when needed. Active exoskeletons are intended to augment human performance by dynamic adjustments of joint motions.

Interaction torque can be used to understand how loads are transferred. Torque metrics are based on sensors which provide measurements of normal forces. Interaction torque is typically taken at the joint to compute the torque transferred from the exoskeleton’s interface to the user [66,93].

Ergonomic metrics, as shown in Table 4, include physical exposure related to task execution. For lifting tasks, the revised National Institute for Occupational Safety and Health (NIOSH) lifting equation provides a set of variables including hand position and posture asymmetry in addition to the task metrics (as shown in Table 5 [69]) in order to evaluate the physical stress of two-handed manual lifting tasks. Wearing exoskeletons can add weight and unexpected forces, which compromise proper posture and balance control. Measures for posture and balance control include the magnitude and velocity of postural kinematics. In addition, interaction forces between a user and the exoskeleton can cause discomfort or pain.

The center-of-pressure (CoP) displacement, in terms of range and speed, over the course of the task provides an understanding of the exoskeleton user’s balance. The anterior–posterior and mediolateral CoP displacements should ideally be similar with and without exoskeleton use or improved with exoskeleton use.

Interaction forces between the physical interface and the user can also impact exoskeleton ergonomics and comfort. Measurement methods include pressure sensors applied.

The performance metrics, shown in Table 5, describe a user’s ability to perform the industrial task while wearing an exoskeleton. These metrics include productivity, load, and quality of the work performed. Motion constraints of exoskeletons can also limit their usability. Constraints can be measured as reduced movement velocities and range of motion (ROM).

The acceptance and adoption of industrial exoskeletons rely on their performance and usability. Both qualitative and quantitative methods to measure usability are provided in Table 6. The cognitive load can be evaluated using the National Aeronautics and Space Administration’s (NASA’s) Task Load Index (TLX) [83], and the cognitive load may be reduced with adequate training. Increased adaptation has led to improved exoskeleton benefits to the user [90]. Quantitative methods to evaluate cognitive load include neuroimaging methods such as functional near infrared spectroscopy (fNIRS)-based brain connectivity [82]. Other usability metrics include donning and doffing speed and ease, measured and perceived comfort [94], human–exoskeleton alignment, fluency, and safety measures such as fall risk. Human–exoskeleton misalignments can be due to drift between the exoskeleton and the user due to the user’s anthropometric measurements, motion, and initial fit [67]. Exoskeleton efficiency and user comfort can be improved through device kinematics and physical interface stability, with the introduction of both rigid and soft components. Further studies of usability metrics can potentially improve exoskeleton alignment to human joints leading to more effective and safer support structures.

## 4. Discussion

The objective of evaluating exoskeletons is to determine whether an exoskeleton meets the user requirements given a specific use case. The process of evaluating an exoskeleton for an intended task includes identifying the use case, selecting evaluation methods based on task type and mobility requirements, identifying metrics aligned with the user requirements, and selecting test subjects aligned with the intended user population. This is then followed by laboratory testing and field testing (Figure 2).

In vivo industrial-exoskeleton evaluations can provide a comprehensive understanding of the different metrics that are relevant to understanding human–exoskeleton interactions. To achieve the highest fidelity, often the test subjects with occupations similar to the intended tasks are selected for the evaluation methods.

### 4.1. Evaluation Challenges

In evaluating exoskeleton performance, the simulated environment’s fidelity to practical industrial and mobility scenarios can vary due to the unique and often dynamic nature of industrial settings. Exoskeleton evaluations can be limited in fully representing the population of exoskeleton users. The limitations can include understanding the exoskeleton’s impact on users with different user-specific health and mobility conditions [68]. As a step towards improving intended user-population representation, the test method fidelity can be subjectively assessed directly by expert workers who would benefit from using the exoskeleton for specific industrial tasks across a wide range of industrial environments.

A survey of laboratory studies resulted in varied experimental designs for evaluating back exoskeletons [8,95]. Standard test methods allow for anyone who fits within the exoskeleton specification to perform the test based on a set of baseline procedures, which can be further developed by the test requestor and tailored to their exoskeleton use case. Variability between users can be reduced through careful selection and representation of test subjects who have had experience and training to perform the industrial task and would be able to benefit from the exoskeleton. Subjects with a similar size and capabilities may also further reduce variability. A survey of laboratory studies resulted in varied experimental designs for evaluating back exoskeletons [8,95]. Standard test methods allow for anyone who fits within the exoskeleton specification to perform the test based on a set of baseline procedures, which can be further developed by the test requestor and tailored to their exoskeleton use case. Variability between users can be reduced through careful selection and representation of test subjects who have had experience and training to perform the industrial task and would be able to benefit from the exoskeleton. Subjects with a similar size and capabilities may also further reduce variability.

Providing appropriate test environmental contexts can also impact exoskeleton performance results. Ideally, the evaluation method would involve expert workers performing their jobs with the exoskeleton in their typical work context. However, field studies can be costly and disruptive to industrial operations, which leads to small and non-representative samples. The studies also require instrumentation to evaluate kinematic, kinetic, and physiological metrics to help improve exoskeleton design. Therefore, both laboratory and field studies can benefit from advancements in low-profile, low-cost, accurate and repeatable measurement capabilities.

Additional standard evaluation frameworks to assess exoskeleton fit to user and to assess user proficiency may also be beneficial in the effective and safe adoption of exoskeleton technologies. Improper exoskeleton fit to user can cause discomfort, reduced mobility, and increased safety risks.The availability of exoskeleton training time can also improve exoskeleton usability, safety, and effectiveness. Moderate training or adaptation periods from 18 to 90 min have been shown to improve exoskeleton benefits [90]. However, cost considerations can also limit the number of subjects, exoskeleton training time, the type and duration of the test, and the types of instrumentation available for exoskeleton evaluation. Other considerations include the potential side effects, adverse events, and longitudinal effects of exoskeleton use. Currently, reported side effects include discomforts and chafing, and may affect users differently based on gender, physical fitness, and age [96]. Side effects are foreseeable effects due to biomechanical or pyschosocial principles, whereas adverse events are unforeseen effects of exoskeleton use [96]. Systematic evaluation methods to understand the side effects and adverse events such as potential balance and posture changes due to improper fit, use, or design can further elucidate potential improvements in exoskeleton design and user training. Another challenge in exoskeleton performance evaluation is understanding the long-term effects of exoskeleton use. Longitudinal studies based on consistent evaluation methods or frameworks remain a needed area of research.

### 4.2. Measurement Challenges

Several measurement challenges exist for the in vivo evaluation of industrial exoskeletons. Human measurements have both inter- and intra-subject variability which can make repeatable measurements difficult to achieve or of higher uncertainty than for in vitro and in-silico evaluation methods. Variations in mood and energy can also affect responses to the use of exoskeletons, to the completion of the task, and to the subsequent survey questions. Sensor measurement variability can occur due to movement, communication, and interference. Variability in test administrator, those who place the sensors on the user, can further contribute to the measurement uncertainty.

The availability of quantitative measurements needed to fully understand the joint loads, load redistribution, and load compensation are also limited [68]. Capturing joint angles consistently using OTS and IMUs can also be a challenge. Both markers and IMU components can drift due to soft-tissue artifacts and stability of the components while the user is active in completing a set of tasks. Furthermore, accurate joint angles require biomechanical modeling tools. Biomechanical simulations can also apply inverse kinematics to estimate joint loads. However, these models are also subject to fidelity and representation limitations.

Managing inter-human variations using algorithmic methods remains an active area of research. The variations in subjective responses can make it difficult to determine the overall effectiveness of the exoskeletons. Responses to wearing exoskeletons do not necessarily correlate with typical demographic categories and may be subject to nuances in individual biodynamics. For example, human–exoskeleton alignment can be due to drift relative to the user (due to fit), human kinematics, and exoskeleton kinematics. To manage the variability in fit, exoskeleton evaluations can also be performed in simulation where the measured data (such as a user’s anthropometric measurements), range of motion (ROM), exoskeleton drift relative to the user, and human–exokeleton alignment can be developed into human models in the loop (MIL) [67]. The aggregate measurements forming the human MILs based on subjects performing a variety of industrial tasks can be used to evaluate how robust an exoskeleton’s self-alignment mechanism is to human and task variations.

Measurement instrumentation used during the test can also affect the behavior of the subject. In general, the instrumentation used to measure subjects should preserve natural subject–exoskeleton interactions and minimize cognitive and physical burdens.

### 4.3. Future Research Directions

Extending exoskeleton performance testing from the laboratory to the field is important to provide a comprehensive understanding of the user–exoskeleton system capabilities and limitations. However, in field environments, instrumentation can be limited to low-cost, portable sensors. In addition, minimizing the profile of the instrumentation on the user enhances the test fidelity and preserves the natural kinematics of the user. Therefore, video-based analysis methods and sensors integrated into the exoskeleton or clothing can further improve the test methods.

Compared with an OTS, photogrammetric methods based on videos are markerless, which is both a benefit to the user who can move more naturally, and a limitation in terms of measurement uncertainty. To leverage photogrammetric data, pose estimations can be derived from the framework of rigid body mechanics [14].

Two areas of research include the development of algorithms and training sets to enable tracking of both user and exoskeleton kinematics. Verification and validation methods [97] are also needed to evaluate the uncertainty of 3D joint positions and orientations, also known as the pose estimation of the musculoskeletal joints and segments [56].

## 5. Conclusions

This paper provides a notional framework to evaluate industrial exoskeletons. A review of current standards and standards under development is provided, along with metrics that have been used for exoskeleton evaluation based on functions, ergonomic support, usability, task performance, and user fit.

## Figures and Tables

**Figure 1 sensors-23-05604-f001:**
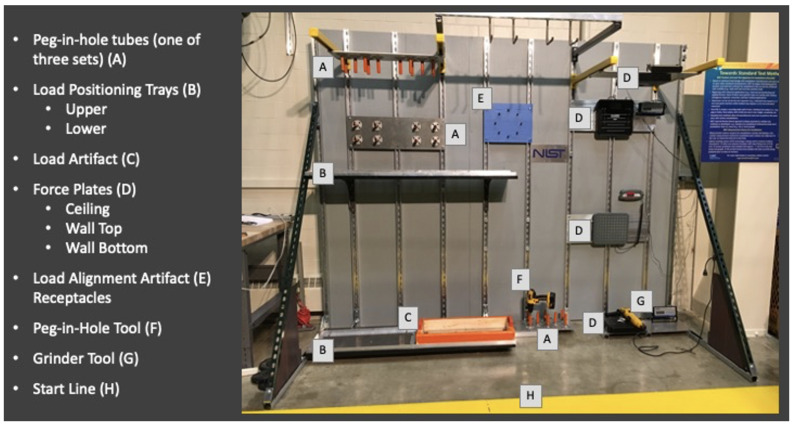
Position and Load Test Apparatus for Exoskeletons (PoLoTAE).

**Figure 2 sensors-23-05604-f002:**
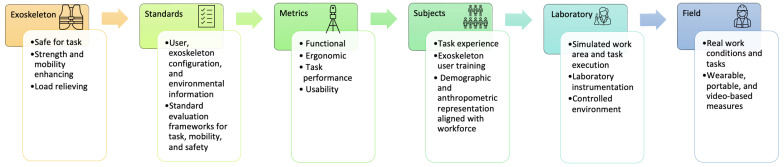
Exoskeleton performance-evaluation-method development process.The process includes selecting an exoskeleton that would be safe and fit for the task; identifying relevant information and exoskeleton evaluation standards framework; identifying relevant metrics; defining test procedures, including integration of measurement methods, for laboratory testing, and, similarly, defining test procedures and measurement methods for field testing.

**Table 1 sensors-23-05604-t001:** Current and proposed industrial-exoskeleton test information standards.

Area	Parameter	Reference
User	Healthy	ASTM F3614 User [23]
	Musculoskeletal Disorders	
	Injuries	
	Occupation	
	Demographic	
	Age	
	Height	
	Weight	
	Profession	
Exo fit to user	User anthropometry	ASTMF3613 Fit [24]
	Configuration	
	Don and doff	
	Range of motion	
Exo configuration	Hardware	ASTM F3576-22 Config [25]
	Hardware settings	
	Control settings	
	User controlled	
	Non-user controlled	
	Power	
Environmental conditions	Indoor or outdoor	ASTM F3427-20 Env Cond [26]
	Humidity	
	Temperature	
	Sanitization	
	Risks	ASTM F3527-21 Env Risks [27]
Training exoskeleton users	Don and doff	ASTM F3444-20 Train Exo [28]
	Fitting	
	Intended use	
	Maintenance and operations	
Metrics	Ergonomic parameters and test metrics	ASTM F3474-20 [29]
		ASTM F3518-21 [30]
Load/material handling	Lifting	ASTM F3443-20 Load handling [31]
	Static holding	
	Carrying	
	Dragging	
	Endurance	
	Symmetric	
	Asymmetric	
	Alignment	

**Table 2 sensors-23-05604-t002:** Current and proposed industrial-exoskeleton evaluation standards for manufacturing tasks and mobility.

Area	Function	Type	Location	Standard
Task	Load/material handling	Lifting	Overhead	ASTM F3443-20 Load handling [31]
		Static holding	Back	
		Carrying	Shoulder	
		Dragging	Side(s)	
		Endurance	Front	
		Symmetric	Pull/push	
		Asymmetric		
		Alignment		
	Assembly	Kneeling	Forward low	
		Stooping	Forward middle	
	Peg-in-hole	Drill/fasten	Forward high	
			Down	
	Applied force	Sanding	Hands	
		Overhead painting	Shoulder	
Mobility	Movement	Sit to stand	Full body	ASTM F3517-21 Movement [32]
	(loaded/unloaded)	Gait		ASTM F3528-21 Gait [33]
		Variable terrain		ASTM WK75742 Terrains [38]
		Inclined plane		ASTM WK84258 Incl Plane [39]
		Stairs		ASTM WK76431 Stairs [40]
		Crawling		ASTM WK83509 Crawling [41]
		Transitions		ASTM WK76543 Transitions [42]
		Ingress/egress		
	Climbing	Ladders		ASTM WK84262 Ladders [43]
		Trees		
	Agility	Confined spaces: vertical		ASTM WK81267 Vertical [44]
		Confined spaces: horizontal		ASTM F3523-21 Horizontal [34]
		Obstacle avoidance		ASTM F3584-22 Obst avoid [45]
		Gaps		ASTM F3582-22 Gaps [36]
		Beams		ASTM F3583-22 Beams [35]
		Hurdles		ASTM F3581-22 Hurdles [37]
	Dexterity	Hand dexterity	Hands	
Safety	Fall risk	Stumbling		ASTM F3578-22 Fall risk [46]

**Table 3 sensors-23-05604-t003:** Industrial exoskeleton functional metrics.

Functional Measure	Metric	Sensors	References
Degrees of freedom (DoF)	A-DOF (assisted DOF)		Pesenti 2021 [61]
	N-DOF (non-actuated DOF)		
Kinematic structure	Rigid		Crea 2021 [8]
	Anthropomorphic		
	Non-anthropomorphic		
	Exosuit (soft)		
Actuation type	Passive		Lowe 2019 [14]
	Quasi-passive		
	Quasi-active		
	Active		
Metabolic cost	*Locomotory economy*		Walsh 2007 [62]
	Walking metabolic power		
	Carbon-dioxide production		Walsh 2007 [62]
	Oxygen consumption	Oxygen mask	Sawicki and Ferris 2008 [63]
	Heart rate	Heart-rate monitor (HRM)	
	Respiratory rate	Pulse oximeter	Wu 2022 [64]
	Skin temperature	Skin temperature sensor	
	Performance index		
	Augmentation factor		Mooney et al., 2014 [65]
Support	*Intended and perceived areas*		Bostelman et al., 2019 [16]
	Back		
	Hand		
	Shoulder		
	Legs		
	Interaction torque	Torque sensor	Massardi 2022 [66]
Reachability	Joint angle	IMU	Sposito 2020 [67]
Capability and dexterity	Joint angle, speed	IMU	Porges 2015 [68]

**Table 4 sensors-23-05604-t004:** Industrial exoskeleton ergonomic metrics.

Ergonomic Measure	Metric	Sensors	References
Hand position	Horizontal location of hands	Video	Waters et al., 1994 [69]
	Vertical location of hands	Video	
		OTS	
	Type of grasp (coupling classification)	Video	
Muscle activity	Reduction	EMG	De Looze et al., 2016 [48]
	Augmentation		ASTM 3518-21 [30]
	Fatigue		Theurel and Desbrosses 2019 [22]
	Strain		Fritzsche et al., 2021 [70]
Joint angle	Knee flexion	OTS	Bostelman et al., 2022 [18]
	Back rotation	IMU	
	Back flexion	Depth perception camera	Yoon et al., 2022 [55]
	Trunk flexion	Monocular video	Rahman 2023 [20]
	Arm/shoulder flexion	Stereo video	
Interaction force	Force between user and exoskeleton	Force Plates	Howard et al., 2020 [5]
		Deep-tissue oxygenation	Kermavnar 2020 [71]
		Pressure sensors	Kermavnar et al., 2018 [72]
			Huysamen et al., 2018 [73]
		Algometry: first-discomfort threshold (FDT), first-pain threshold (FPT), time to FDT	Kozinc et al. 2021 [74]
	Normal, shear		
Posture	Postural control	Force plates	Kim et al., 2018 [75]
	Asymmetry	OTS, video (pose estimation)	Waters et al., 1994 [69]
	CoP displacement	Pressure walkway	Wu et al., 2022 [64]
Balance	CoP displacement	Pressure insoles	
	Anterior–posterior (AP)		
	Mediolateral		
Kinematics	Human–exo alignment	OTS	Bostelman et al., 2022 [18]

**Table 5 sensors-23-05604-t005:** Industrial exoskeleton task performance metrics.

Task Measure	Metric	Method	References
Productivity	Perceived exhaustion	Borg scale	CDC [76]
		Likert scale	Gob 2007 [77]
	Task rate	Video (pose estimation)	Bostelman 2022 [18]
	Task completion time	Video (manual counting)	Kim et al., 2018 [10]
	Number of errors		
	Task duration		Bostelman, et al., 2018 [78]
			Bosch 2016 [79]
Load	Lifting index		Waters 1994 [69]
	Composite lifting index		
	Lift duration	Video	
	Lift frequency	Video	
	Travel distance	Video	
	Load weight		
Mobility	Movement velocity	IMU, OTS	Otten 2018 [80]
		Video (pose estimation)	
Quality	Task precision	Video (pose estimation)	Howard 2020 [5]
			Bostelman 2018 [78]
Range of motion	Task angle	OTS	Kim et al., 2018 [10]
	Task height		Kim et al., 2018 [75]
		Video (pose estimation)	Yoon et al., 2022 [55]
Mobility	Gait speed	Pressure walkway	Kim et al., 2018 [75]
	Stride length	Video (pose estimation)	Yoon et al., 2022 [55]
			Baltrusch et al., 2019 [81]

**Table 6 sensors-23-05604-t006:** Industrial exoskeleton usability.

Task Measure	Metric	Method	References
Complexity	*Cognitive load*	fNIRS-based brain connectivity	Zhu 2021 [82]
	Mental demand	TLX scale	NASA-TLX 2006 [83]
	Physical demand		
	Temporal demand		
	Performance		
	Effort		
	Frustration		
	Visual misses	Number missed (visual)	Bequette 2020 [84]
	Visual reaction time	Time: visual cue and reaction	
	Audio misses	Number missed (audio)	
	Audio reaction time	Time: audio cue and reaction	
	Lag time		
	Don/doff time	Time to wear and to remove without aid	
	Don/doff ease	Likert scale	
Comfort	Perceived	Likert scale	De Looze et al., 2016 [48]
	Thermal comfort	Thermal ratings (comfort, acceptability, sensation)	Elstub et al., 2021 [85]
		Skin temperature sensor	
	Pain, pressure, discomfort	Body location and severity	Bostelman et al., 2019 [16]
			Han et al., 2022 [86]
		Deep-tissue oxygenation	Kermavnar 2020 [71]
		Pressure sensors	Huysamen et al., 2018 [73]
		Algometry: first-discomfort threshold (FDT), first-pain threshold (FPT), time to FDT	Kozinc et al. 2021 [74]
	Chafing	Pressure sensor	DeRossi 2010 [87]
Exertion	Subjective effort	Likert scale	Theurel and Desbrosses 2019 [22]
		Borg CR10	Otten et al., 2018 [80]
Fit	Human–exo alignment	OTS	Bostelman 2022 [18]
	Comfort	Likert scale	
	Pressure	Pressure measurement over contact area	DeRossi 2010 [87]
	Drift	IMU	Sposito et al., 2020 [67]
Task difficulty	Subjective effort	Likert scale	Theurel and Desbrosses 2019 [88]
Acceptance	Satisfaction	Likert scale	Bostelman, et al., 2019 [16]
Fluency	Joint moment similarity		Kao 2010 [89]
	Human–exo concurrency		Baltrusch 2022 [11]
	Idle time		
	Training and use hours		Poggensee 2022 [90]
Fall risk		Force plates	Kim et al., 2018 [75]
			Luger et al., 2019 [91]

## Data Availability

Publicly archived datasets for the NIST exoskeleton test method development studies can be found at: https://www.nist.gov/el/intelligent-systems-division-73500/exoskeletons-and-exosuits-research-and-standard-test-methods-1.

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
