# Peer review of "Evaluation Methods and Measurement Challenges for Industrial Exoskeletons"

_sensors, 2023, doi:10.3390/s23125604_

Round 1

Reviewer 1 Report

This paper discusses the current state of evaluation methods for industrial exoskeletons. These methods include both laboratory and field simulations, and they use various physiological, kinematic, kinetic, and subjective metrics to evaluate the usability, safety, and effectiveness of exoskeletons in reducing musculoskeletal injuries. The paper proposes a classification of these metrics based on their relevance to exoskeleton fit, task efficiency, comfort, mobility, and balance.

While exoskeletons hold promise for reducing musculoskeletal injuries and improving worker productivity, there are several potential drawbacks associated with their use.

One drawback is the cost of exoskeletons, which can be quite expensive, making them inaccessible for smaller businesses or individual workers. Additionally, some exoskeletons may not fit all workers properly, which can lead to discomfort, reduced mobility, and even injury.

Another potential drawback is the need for workers to be trained in the proper use of the exoskeleton, which can be time-consuming and may require additional resources. In some cases, the use of exoskeletons may also lead to workers becoming over-reliant on the technology, which could impact their physical fitness and lead to decreased strength over time.

Finally, there is a need for further research into the long-term effects of exoskeleton use, particularly in terms of their impact on worker health and safety over time. While exoskeletons have shown promise in reducing musculoskeletal injuries in the short-term, it is unclear whether they will have the same benefits over a longer period of time.

Author Response

Thank you for the careful review and insightful suggestions. All of the points are well taken. Instead of the Introduction section, we thought the points may be better addressed in the Discussion section, Evaluation Challenges.  We noted the drawbacks and limitations of exoskeletons as well as the need for longitudinal studies, including several references discussing the side effects and long-term effects. The cost factor is a consideration in the acceptance of the exoskeleton for adoption by a user or small business; however, we consider it outside the scope of this study.

We added the following to the Discussion section:

“Additional standard evaluation frameworks to assess exoskeleton fit to user and to assess user proficiency may also be beneficial in the effective and safe adoption of exoskeleton technologies.  Improper exoskeleton fit to user can cause discomfort, reduced mobility, and increased safety risks.The availability of exoskeleton training time can also improve exoskeleton usability, safety, and effectiveness. Moderate training or adaptation periods from 18 to 90 minutes have been shown to improve exoskeleton benefits [75]. However, cost considerations can also limit the number of subjects, exoskeleton training time, the type and duration of the test, and the types of instrumentation available for exoskeleton evaluation. Other considerations include the potential side effects, adverse events, and longitudinal effects of exoskeleton use. Currently reported side effects include discomforts and chafing, and may affect users differently based on gender, physical fitness, and age [94]. Side effects are foreseeable effects due to biomechanical or pyschosocial principles, whereas adverse events are unforeseen effects of exoskeleton use [94]. Systematic evaluation methods to understand the side effects and adverse events such as potential balance, posture, and muscle activity changes due to exoskeleton use or improper fit can further elucidate potential improvements in exoskeleton design and user training. Longitudinal studies based on consistent evaluation methods or frameworks remain a needed area of research.”

Reviewer 2 Report

Thank you for the broad and excellent view on science, needs, tasks, measurements etc. on exoskeletons.

The remaining question is whether the optimal exoskeleton can be produced for the high variety of demands and the different individual needs.

1) Topic

The paper is a survey about measurement methods. These are discussed from different points of view (e.g. simulation, in vivo, task specific etc., sorts of exos etc. and a final view in the future of the use of exos.

2) Relevance

In order to get a deeper understanding many aspects of exos should be elucidated. Exactly this is done by the authors. This includes a consideration of older and actual sources, which are delivered from the authors.

3) Subject area

The broadness of this approach is the main benefit. If s.th. is done this way it automatically cannot elucidate any details of research e.g. on exclusively back exos.

4 and 7)Improvement/controls/tables

From my point of view the authors succeed in presenting their topic. The tables are somewhat long and this could be put in a new, i.e. better readable design.

5) Conclusions

The authors have a clear structure and a clear aim. The conclusions are diligently based on investigations from literature with different methods and from my perspective are appropriate.

6) The references include a broad spectrum of sources.

Author Response

Thank you for the kind review. We have continued to improve on clarifying and editing the text including the addition of several standards, updating the figures, and providing more explanations on the measurement methods, differences, and tradeoffs. We have also improved on the graphics and discussions of the evaluation challenges.

Reviewer 3 Report

The paper discusses various methods of evaluating and measuring industrial exoskeletons, providing the reader with a comprehensive overview of standards and literature references. In effect, this is a review article. It might be appropriate to change the title by explicitly using the word "review."

Although the innovative and scientific content is limited, the nature of the paper is a survey of the state of the art, which is done thoroughly and comprehensively. The overall presentation is good.

Author Response

We considered adding “review”, and prefer to keep the current title, if it is acceptable. The work does review what is available in literature, but also (1) enumerates standard test methods and information standards; (2) discusses the challenges in evaluating exoskeletons in literature; and (3) proposes a process for exoskeleton evaluation to mitigate some of the limitations / challenges.  

Reviewer 4 Report

This is a well written article and summarized the current standard development for industrial exoskeletons. It will benefit the researchers in this field. This reviewer has the following suggestions. 

1. Authors presented many qualitative descriptions about the standard development. It would be desirable to provide some quantitative standard values. 

2. Authors focused on lab or field experimental methods and missed recent developments of simulation methods for exoskeleton evaluations. 

3. ASTM released the first industrial exoskeleton evaluation standards this year, please try to include some details of this recent standard. 

4. Authors may present some related exoskeleton products on the market and summarize their applications and shortcomings. 

Author Response

  1. Authors presented many qualitative descriptions about the standard development. It would be desirable to provide some quantitative standard values. 

If I understand correctly, we have covered quantitative and qualitative measures described in standards. We expanded on the objectives and challenges of these quantitative measures including the use of OTS, video, and IMUs.

  1. Authors focused on lab or field experimental methods and missed recent developments of simulation methods for exoskeleton evaluations. 

As noted in the introductory around line 40:

“The types of exoskeleton evaluation methods can be categorized as (a) in vitro, where
evaluation is performed in a controlled environment on a physical model, such as a
mannequin; (b) in vivo, where evaluation is performed with human test subjects either in a
laboratory environment or an industrial environment; and (c) in silico, where computational
simulations are applied to analyze human-exoskeleton interactions
[ 9 ]. This study focuses
on in vivo exoskeleton test methods.”

  1. ASTM released the first industrial exoskeleton evaluation standards this year, please try to include some details of this recent standard. 

We re-reviewed the standards from the F48 committee. The evaluation standards for fall risk from 2022 was added. The other evaluations standards, such as mobility, from 2022 have been included. We added additional information standards related to the training (ASTM 3444-2020).

The following text was added to expand on the details and use of the recent standards:

Table 2 extends the contextual information to include evaluation of exoskeletons for 137
specific industrial tasks. These evaluation standards are categorized into task-based and 138
mobility-based evaluation standards. The task-based evaluation standards and proposed 139
standards include load handling [ 20 ] and applied force tasks such as grinding, sanding, and overhead painting [ 21 ]. Several exoskeleton performance standards intended to evaluate the effects of exoskeletons on the users mobility include sit-to-stand [22], gait[ 23], horizontal confined spaces [ 31 ], beams [ 39], gaps[ 38 ], and hurdles[40 ]. The exoskeleton evaluation standards provide a framework in which an evaluator can decide on the specific environment, measurements, test procedures, and information collection based on the intended task and the intended use of the exoskeleton. For both safety and effectiveness of using an exoskeleton, evaluators can consult with the exoskeleton manufacturer to determine whether the test protocol follows proper exoskeleton maintenance, operations, and safety risk management. In addition, the necessary training for both the evaluators and the test subjects are sufficient to safely operate the exoskeleton and follow the test protocol.”

  1. Authors may present some related exoskeleton products on the market and summarize their applications and shortcomings.

NIST has a policy to remain neutral about products and it is out of our scope to evaluate exoskeleton products. However, we can develop and evaluate test methods.

Reviewer 5 Report

The paper describes the state-of-the-art measurement methods applied in evaluating industrial exoskeletons. It discusses the physiological, kinematic, and kinetic metrics and subjective surveys used to assess exoskeleton usability, fit, comfort, mobility, and balance. The paper proposes a classification of metrics based on these factors to guide potential exoskeleton users in selecting appropriate standard test methods and metrics for their industrial tasks. The article also details the test and measurement methods used in supporting the development of exoskeleton evaluation methods.

The paper is well-written, and it may interest the community. Although the paper has potential, there are significant concerns about detailing the methods and minor issues to address.

Major Issues:

-The main concern is on the methods detailed. There are few or no explanations of the methods; therefore, the paper must be enriched with information to give a direction to the reader. For example, some items that could have a better description:

- What is the goal of measuring EMG in comparison to other methods? The only mention in the text is that one should measure EMG with force measurement for the back muscle strain test. What are other uses? 

- What is the difference between OTS and other video methods? Furthermore, what are the marker models? 

- IMU appears as a metric in the table. 

- What is a Locomotory economy?

These are some examples of terms and methods that authors should explain better, but this extends for the rest of the paper.

The review is good in length, but all items should be explained and discussed in the text. 

Minor issues:

- Acronyms should be defined at first appearance. 

- Figure 1 arrows are hard to read. Use letters or numbers on the Figure and identify them in the text or legend. 

- Figure 2 seems off. Blank background? Is the Figure necessary? It would be more useful if the authors included more information in the Figure. A suggestion is to include subitems in each process step with the variations discussed in the text;

Author Response

Dear Reviewer,

Thank you for your helpful feedback to improve the document. Please see below the point-to-point responses to address the issues. 

Major Issues:

  • The main concern is on the methods detailed. There are few or no explanations of the methods; therefore, the paper must be enriched with information to give a direction to the reader. For example, some items that could have a better description:
    • What is the goal of measuring EMG in comparison to other methods? The only mention in the text is that one should measure EMG with force measurement for the back muscle strain test. What are other uses? 

The following text was added:

“EMG sensors, for the neck, back, shoulder, arm, and leg muscles, have been applied to track muscle activity. EMG signals provide quantitative measures to better understand the impact of an exoskeleton on physiological exertion. EMG signals can track the magnitude as well as the temporal and spatial properties of the physiological changes introduced with the use of exoskeletons [ 56 ]. The amplitude of the EMG signal describes the peak muscle activity. Typically, a decrease in peak muscle activity is expected with the use of exoskeletons. An increase in peak muscle activity can imply the transfer of loads from one muscle region to another. A decrease in mean power frequency of EMG signals has also been shown to indicate neuromuscular fatigue [84][85 ] and to indicate the point before and after the force begins to decline, which is also known as the endurance point [86].”

  • What is the difference between OTS and other video methods? Furthermore, what are the marker models? 

The main difference between OTS and video is that OTS is marker-based and video would be markerless. The following text was added:

“Videos and OTS can both be used to track user kinematics for human motion analysis. An OTS can be used by modifying the marker models to manage occlusions. The marker models are a set of OTS markers placed precisely on anatomical landmarks to track the exoskeleton wearer’s motion. The tracking software can include baseline and conventional marker sets [87]. The advantage of using clinically validated marker models, such as the Helen Hayes marker set [88], is that the measured data can be used by biomechanical analysis software to derive more repeatable and accurate joint angle measurements than non-validated models and potentially video-based methods as well. In addition, the kinematic data from tracking the exoskeleton wearer’s marker set, can be combined with kinetic measurements, such as force plates, to compute joint moment profiles [88]. The challenge in using an OTS to track an exoskeleton wearer’s motion is primarily marker occlusion. To overcome occlusion, slight modifications to the marker model have been applied to track the wearer’s motion. Another method is to use common artifacts with a set of markers both on the exoskeleton and on the wearer to synchronously track the kinematic alignment [39 ]. Markerless human motion analysis methods, using video or depth-perception cameras, rely on stereophotogrammetric methods to estimate bone position, bone orientation, and joints within a rigid body framework [74 ][42 ]. Video-based methods have the advantage of being portable, low-cost, and readily adaptable for use in field environments.”

  • IMU appears as a metric in the table. 

We corrected and verified that IMU now appears only as a sensor / method for evaluating relevant exoskeleton performance criteria.

  • What is a Locomotory economy?

The following text was added:

Locomotory economy metrics, such as cost of transport (COT), are performance measures evaluating the metabolic cost efficiency of an exoskeleton as the wearer, with or without additional load, moves from one location to another. For example, for exoskeletons designed to augment load-carrying abilities by reducing the work to the wearer, COT can be derived by measuring the rate of oxygen consumption of the subject with and with the exoskeleton [67].

These are some examples of terms and methods that authors should explain better, but this extends for the rest of the paper. The review is good in length, but all items should be explained and discussed in the text. 

Additional terms and methods were explained including IMU, augmentation factor, CoP displacement, interaction torque, and fNIRS. The main idea is to refer the reader to the respective references on the methods; however, we agree it also useful to briefly clarify in the text here.

Minor issues:

  • Acronyms should be defined at first appearance. 

We reviewed the document for acronyms, and defined them according to first appearance.

  • Figure 1 arrows are hard to read. Use letters or numbers on the Figure and identify them in the text or legend. 

We modified the graphic to use letters.

  • Figure 2 seems off. Blank background? Is the Figure necessary? It would be more useful if the authors included more information in the Figure. A suggestion is to include subitems in each process step with the variations discussed in the text;

We modified the process graphic with subitems to capture the process of evaluating an exoskeleton – selecting an exoskeleton that is fit for the task, identify relevant information and evaluation standards framework, identify relevant metrics, define test procedures, including integration of measurement methods, for laboratory testing, and similarly define test procedures and measurement methods for field testing.

Thank you for your time and consideration. 

Sincerely,

Ya-Shian Li-Baboud

Round 2

Reviewer 1 Report

Accept 

Reviewer 5 Report

Thank you for addressing all the revisions requested during the review process. I have carefully reviewed the revised manuscript and am pleased to inform you that the necessary changes have been made satisfactorily. Based on the improvements made, I recommend accepting the paper for publication.